# Glutathione determines chronic myeloid leukemia vulnerability to an inhibitor of CMPK and TMPK
Chang-Yu Huang [ORCID] [1], Yin-Hsuan Chung[1], Sheng-Yang Wu[1], Hsin-Yen Wang[1], Chih-Yu Lin[2], Tsung-Jung Yang[3], Jim-Min Fang[3], Chun-Mei Hu[4] & Zee-Fen Chang [ORCID] [1,5] ✉

Bcr-Abl transformation leads to chronic myeloid leukemia (CML). The acquirement of T315I mutation causes tyrosine kinase inhibitors (TKI) resistance. This study develops a compound, JMF4073, inhibiting thymidylate (TMP) and cytidylate (CMP) kinases, aiming for a new therapy against TKI-resistant CML. In vitro and in vivo treatment of JMF4073 eliminates WT-Bcr-Abl-32D CML cells. However, T315I-Bcr-Abl-32D cells are less vulnerable to JMF4073. Evidence is presented that ATF4-mediated upregulation of GSH causes T315I-Bcr-Abl-32D cells to be less sensitive to JMF4073. Reducing GSH biosynthesis generates replication stress in T315I-Bcr-Abl-32D cells that require dTTP/dCTP synthesis for survival, thus enabling JMF4073 susceptibility. It further shows that the levels of ATF4 and GSH in several human CML blast-crisis cell lines are inversely correlated with JMF4073 sensitivity, and the combinatory treatment of JMF4073 with GSH reducing agent leads to synthetic lethality in these CML blast-crisis lines. Altogether, the investigation indicates an alternative option in CML therapy.

The Philadelphia chromosome, arising from the reciprocal translocation between chromosomes 9 and 22, is identified in chronic myeloid leukemia (CML)[1,2]. This chromosomal translocation causes the fusion of the *ABL* gene on chromosome 9 with the *BCR* gene on chromosome 22, giving rise to the *BCR-ABL* fusion gene. *BCR-ABL* encodes a constitutively active tyrosine kinase that phosphorylates a panel of downstream targets to cause uncontrolled cell proliferation, protein synthesis, and anti-apoptotic signals[3–6]. Although Bcr-Abl inhibitors such as imatinib, nilotinib, and dasatinib are effective in CML therapy[7–9], the acquisition of the T315I mutation of *BCR-ABL* in patients causes the resistance to most tyrosine kinase inhibitors (TKI), leading to poor outcomes. A structure-based designed inhibitor of Bcr-Abl-T315I, ponatinib, was developed and used in the clinical trial. However, the side effects were severe due to cardiotoxicity[7,10,11]. In addition to the acquirement of drug-resistant mutation, the development of cancer stem cells also causes CML persistence and relapse from TKI therapy[12–14].

Uncontrolled proliferation associated with oncogene overexpression often exhausts dNTP pools, leading to replication fork stalling, double-strand breaks, and genome instability[15,16]. It has been established that the speed of replication fork progression is regulated by TIPIN/TIMELESS complex with oligomer form of peroxiredoxin 2 (PRDX2), as a ROS sensor, in replisome. Elevated ROS causes the monomeric formation of PRDX2.

This leads to the dissociation of TIMELESS from the replisome, slowing replication and resulting in replication stress and genome instability[17]. Therefore, oxidative stress also determines oncogene-induced replication stress. Since ROS-dependent DNA damage is markedly increased in myeloid cells after Bcr-Abl transformation[18], we hypothesize that blocking dTTP and dCTP supply might exacerbate replication stress and DNA damage in Bcr-Abl-transformed cells, thereby jeopardizing cell survival. Herein, this study aims to develop an inhibitor to reduce dTTP and dCTP pools without general toxicity as an option for CML treatment.

We have previously identified a compound YMU1, an inhibitor of thymidylate kinase (TMPK), which increased doxorubicin sensitivity of cancer cells by causing DNA repair toxicity[19,20]. In this study, we developed a new molecule JMF4073, which inhibits both thymidylate kinase (TMPK) and cytidylate kinase (CMPK). We showed that, unlike other conventional anti-cancer agents, 5-fluorouracil (5-FU) or cytosine arabinoside (ara-C), which are uracil and cytidine analogs, JMF4073 is not toxic to untransformed myeloid cells. This study found that JMF4073 treatment suppresses the in vitro and in vivo growth of WT-Bcr-Abl-transformed leukemia cells, but T315I-Bcr-Abl-transformed cells are less susceptible. WT- and T315I-Bcr-Abl-transformed cells are very different in replication stress, dTTP, and glutathione (GSH) levels. The RNA sequencing data lead us to find that

[1]Institute of Molecular Medicine, College of Medicine, National Taiwan University, Taipei, Taiwan. [2]Agricultural Biotechnology Research Center, Academia Sinica, Taipei, Taiwan. [3]Institute of Chemistry, National Taiwan University, Taipei, Taiwan. [4]Genomics Research Center, Academia Sinica, Taipei, Taiwan. [5]Center of Precision Medicine, College of Medicine, National Taiwan University, Taipei, Taiwan. ✉e-mail: zfchang@ntu.edu.tw

ATF4-mediated gene transcription network[21–23] is more active in T315I-Bcr-Abl cells. This network upregulates the expression of serine hydroxymethyltransferase-2 (*SHMT2*) for dTTP synthesis and amino acid transporters, such as solute carrier family 1 member 1 (*SLC1A1*) and solute carrier family 7 member 8 (*SLC7A8*), for GSH biosynthesis.

Our further investigation reveals that the upregulation of GSH bio-synthesis rather than dTTP is responsible for avoiding replication stress in T315I-Bcr-Abl cells. Blocking mitochondrial pyruvate carriers by UK-5099 in T315I-Bcr-Abl cells reduces GSH level to induce replication stress. The combination of UK-5099 with JMF4073 depleted dTTP and dCTP pools to achieve both in vitro and in vivo therapeutic effects. Moreover, we validated the relationship between GSH-regulated replication stress and JMF4073 sensitivity in a number of TKI-insensitive human CML blast-crisis cells, and present the evidence of the synthetic lethality by targeting glu-tathione biosynthesis with JMF4073 in these cells.

## Results

### Development of JMF4073 as an inhibitor of TMPK and CMPK

The survival and growth of cancer cells require a sufficient supply of the four dNTPs for DNA repair and replication. Thymidylate kinase (TMPK) and cytidylate kinase (CMPK) are essential for the biosynthesis of TTP and the salvage synthesis of dCTP, respectively (Fig. 1a). 5-Fluorouracil (5-FU), which irreversibly inhibits thymidylate synthase, has been widely used for chemotherapy. However, 5-FU exerts general toxicity on normal pro-liferating cells due to 5-FdUTP and dUTP misincorporation into DNA and 5-FUTP into RNA[24]. We have previously identified a TMPK inhibitor, namely YMU1. Using YMU1 as a lead, the compounds JMF2977 and JMF4073 were synthesized. The IC50 of YMU1, JMF2977, and JMF4073 was 0.8, 0.5, and 0.16 μM, respectively. JMF4073, containing a fluoride substitution at the C-position of the pyridine ring, has the lowest IC50 value and higher solubility, as indicated by the clogP value (Fig. 1b). By molecular simulation, we have previously shown that YMU1 acts on the catalytic pocket of TMPK, which causes the expulsion of the lid domain to give an open conformation that inhibits the catalytic process[19]. Since CMPK also has a lid domain[25], we then tested whether JMF4073 is an inhibitor of CMPK. The results showed that JMF4073 inhibits both TMPK and CMPK at a similar IC50, 0.17 μM, whereas YMU1 is less potent to CMPK (Fig. 1b). Thus, JMF4073 is indeed an inhibitor of TMPK and

CMPK. In accordance, treatment of mouse 32D myeloid progenitor and human HEK-293T cells with JMF4073 for 6 h significantly reduced the cellular level of TTP and dCTP (Fig. 1c).

We then determined the mode of JMF4073 inhibition by pre-incubating purified TMPK or CMPK protein with various concentrations of JMF4073 and measured the alteration in Vmax and *Km* by NADH-coupled enzymatic assay. The kinetic data demonstrated that JMF4073 inhibits TMPK and CMPK in a mixed inhibitory mode as indicated by the increased *Km* for the substrates with decreased Vmax. The inhibition constant (Ki) of JMF4073 in the dTMP and ATP kinetic analysis of TMPK was 0.16 μM and 0.005 μM, respectively. The Ki in the CMP and ATP kinetic analysis of CMPK was 0.37 μM and 0.06 μM, respectively (Table 1). These kinetic analyses indicate that JMF4073 has a higher affinity to the ATP binding sites in TMPK and CMPK.

### JMF4073 eliminates WT-Bcr-Abl-transformed myeloid cells in vitro and in vivo

Since the balanced supply of dNTPs is critical for overcoming DNA damage and replication stress for cell survival, we then tested whether JMF4073 treatment could selectively suppress the growth of Bcr-Abl-transformed but not untransformed myeloid cells. To this end, 32D cells, a mouse myeloid progenitor line which proliferation is dependent on interleukin-3 (IL-3)[26], were employed for WT- and T315I-Bcr-Abl transformation. The pro-liferation of these transformed cells was independent of IL-3[26]. By examining a variety of Bcr-Abl downstream signals including pAkt-S473, pS6K-T389 in the PI3K/Akt/mTOR pathway, and pStat5-Y694, we confirmed these signals similarly elevated in these two different transformed cells. As expected, these signals were sensitive to imatinib treatment only in WT-Bcr-Abl-32D cells but not T315I-Bcr-Abl-32D cells, while ponatinib[27] treatment abolished these signals in both cell lines (Supplementary Fig. 1). It was noted that WT- and T315I-Bcr-Abl transformation markedly stimulated the level of overall tyrosine phosphorylation, which was reduced by ponatinib treatment. However, unlike imatinib treatment, the basal level of tyrosine phosphorylation in 32D progenitor cells was also reduced by ponatinib, indicating its non-specific inhibitory effect (Fig. 2a).

Next, JMF4073 sensitivity in non-transformed, WT-, and T315I-Bcr-Abl-transformed 32D cells was compared. The results showed that WT-Bcr-Abl cells were highly susceptible to JMF4073 with GI50 0.67 μM

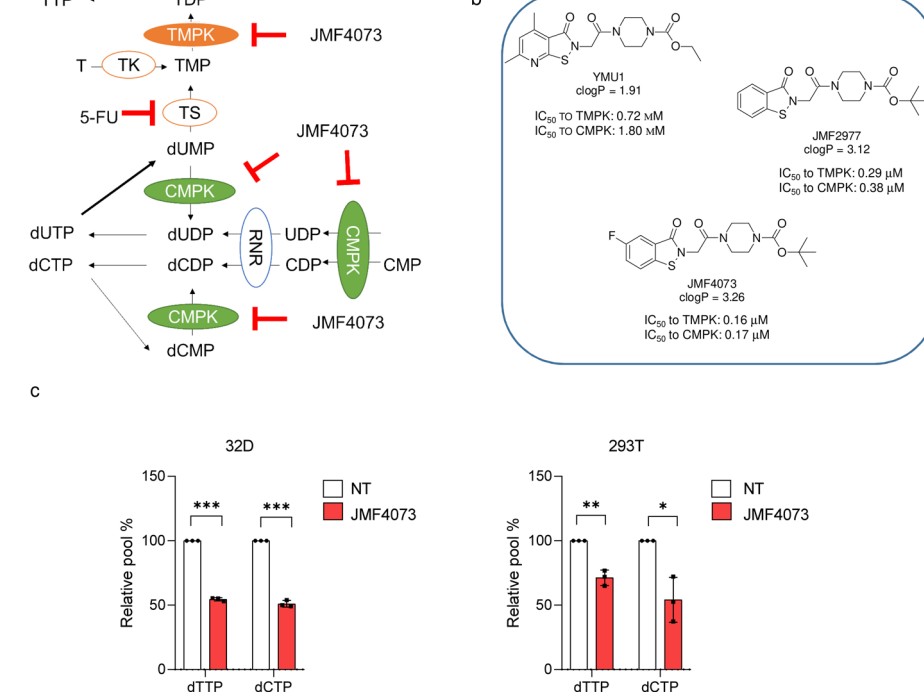

**Fig. 1 | JMF4073 inhibits TMPK and CMPK.**
**a** Schematic view of the synthesis of dTTP and dCTP. TMPK and CMPK mediate the synthesis of dTDP and dCDP, respectively, which subsequently are converted to dTTP and dCTP by nucleoside diphosphate kinase (NDPK). Thymidylate synthase (TS) converts dUMP to dTMP, which is inhibited by 5-FU. Ribonucleotide reductase (RNR) mediates the de novo synthesis of dCDP, dADP, dUDP, and dGDP. Salvage synthesis of dTMP and dCMP is mediated by thymidine kinase and deoxycytidine kinase. **b** The chemical structures, clogP, and IC50 of YMU1, JMF2977, and JMF4073. The clogP values were calculated by Chemsketch. The IC50 values against hTMPK and hCMPK were measured by NADH-coupled TMPK assay. **c** The measurement of dTTP and dCTP pools in untransformed 32D and HEK-293T cells after incubation with JMF4073 (10 μM) for 6 h. Data are represented as means ± S.D., n = 3 biological replicates. Asterisks denote *p < 0.05, **p < 0.01, or ***p < 0.001, from unpaired two-tailed Student's *t*-test.

**Table 1 | Effects of JMF4073 on enzymatic kinetics of purified human TMPK and CMPK**

| TMPK | | | |
|---|---|---|---|
| JMF4073 (µM) | Km for TMP (µM) | Vmax (nmol/min/mg) | Ki (µM) |
| 0 | 14.2 ± 4.04 | 333.3 ± 20.93 | 0.16 |
| 0.1 | 13.4 ± 4.83 | 248.2 ± 19.26 | |
| 0.2 | 18.0 ± 3.93 | 133.2 ± 7.03 | |
| 0.4 | 57.1 ± 16.55 | 114.1 ± 11.94 | |
| JMF4073 (µM) | Km for ATP (µM) | Vmax (nmol/min/mg) | Ki (µM) |
| 0 | 5.1 ± 0.85 | 312.5 ± 9.20 | 0.005 |
| 0.1 | 88.5 ± 26.14 | 318.3 ± 44.33 | |
| 0.2 | 157.4 ± 68.49 | 89.0 ± 22.26 | |
| 0.4 | 171.1 ± 59.27 | 76.0 ± 15.54 | |
| CMPK | | | |
| JMF4073 (µM) | Km for CMP (µM) | Vmax (nmol/min/mg) | Ki (µM) |
| 0 | 120.5 ± 29.04 | 478.3 ± 96.97 | 0.37 |
| 0.1 | 127.4 ± 32.11 | 344.0 ± 44.63 | |
| 0.2 | 165.1 ± 29.88 | 311.2 ± 17.64 | |
| 0.4 | 127.6 ± 35.86 | 174.2 ± 20.92 | |
| JMF4073 (µM) | Km for ATP (µM) | Vmax (nmol/min/mg) | Ki (µM) |
| 0 | 4.6 ± 0.69 | 393.5 ± 13.68 | 0.06 |
| 0.1 | 6.1 ± 1.23 | 367.9 ± 25.91 | |
| 0.2 | 10.0 ± 1.38 | 151.9 ± 8.22 | |
| 0.4 | 628.1 ± 309.4 | 107.6 ± 24.65 | |

Purified hTMPK (0.4 µg) or hCMPK (0.3 µg) were pre-incubated with JMF4073 at the indicated concentration for 10 min, and the initial velocities of hTMPK and hCMPK reaction were measured by NADH-coupled assay. The kinetic assays of hTMPK activity were performed in the reactions containing increasing concentration of TMP (0 to 250 µM) in the presence of 500 µM ATP or increasing concentration of ATP (0 to 200 µM) in the presence of 500 µM of TMP. hCMPK inhibition analyses were performed in the reactions containing increasing concentration of CMP (0 to 500 µM) in the presence of 500 µM ATP or increasing concentration of ATP (0 to 500 µM) in the presence of 500 µM of CMP. Data represent mean ± S.E.M, n = 3. Data obtained from the nonlinear regression analysis were used to calculate the Km and Vmax. The Ki value of JMF4073 for TMPK and CMPK were calculated from the model of non-competitive or mixed inhibition by Prism.

as compared to 1.5 µM and 4.2 µM for T315I-Bcr-Abl-32D and untransformed 32D cells, respectively (Fig. 2b). The levels of overall tyrosine phosphorylation in these cells were unaffected by JMF4073 treatment (Fig. 2c). 5-FU and cytosine arabinoside (ara-C) are nucleoside analog drugs widely used to induce DNA and RNA toxicity for chemotherapy[28,29]. The sensitivity of 5-FU and ara-C were similar in WT-, T315I-Bcr-Abl-32D and untransformed 32D progenitor cells (Fig. 2d). Thus, JMF4073 treatment has an advantage over 5-FU and ara-C in selectively suppressing Bcr-Abl-transformed but not untransformed myeloid progenitor cells.

We further tested the in vivo therapeutic effect of JMF4073 in WT-Bcr-Abl-CML mice. WT-Bcr-Abl-transformed 32D cells were transplanted into C3H/HeNCrNarl mice. After 48 h, these mice were treated with a daily intraperitoneal injection of JMF4073 for 14 days (Fig. 2e). After the therapy, the hematology analysis showed white blood counts were brought down by JMF4073 treatment in WT-Bcr-Abl-32D bearing mice (Supplementary Table 1). Moreover, treatment with JMF4073 for 14 days prolonged the survival of leukemia mice (Fig. 2f). Since T315I-Bcr-Abl-transformed 32D cells took longer time to develop leukemia phenotype than WT Bcr-Abl cells, we transplanted 2-fold more cells into C3H/HeNCrNarl mice. After transplantation for 7 days, these mice were treated with JMF4073 for 14 days. Unlike the WT-Bcr-Abl mice, the survival of T315I leukemia mice was not significantly prolonged by JMF4073 (Fig. 2g; Supplementary Table 2). Thus, unlike WT-Bcr-Abl cells, T315I-Bcr-Abl cells are less vulnerable to JMF4073 in vitro and in vivo.

## T315I-Bcr-Abl-32D cells have high dTTP pool and low replication stress with increased GSH biosynthesis

To understand the underlying mechanism for the differential susceptibility to JMF4073 in these cells, we performed quantitative analysis of four dNTPs. Data showed that WT-Bcr-Abl-32D cells had very low levels of dNTPs as compared to those in T315I-Bcr-Abl-32D and untransformed 32D cells, with the most significant difference in dTTP pool (Fig. 3a). Compelling evidence has shown that oncogene-induced replication stress and dNTP exhaustion[30,31]. Since nucleotide supply plays a critical role in DNA replication, we further performed DNA fiber assay by measuring the lengths of chase IdU-labeled DNA replication tracks that link to pulse CldU-labeling, which is a gold standard for assessing replication stress[32]. The analysis showed that the replication track lengths in WT-Bcr-Abl cells were obviously much shorter as compared to those in 32D cells (Fig. 3b). Intriguingly, the replication track lengths were quite similar in T315I-Bcr-Abl-32D and 32D cells. These results evoke the speculation that higher dTTP level in T315I-Bcr-Abl-32D cells might prevent DNA replication stress to affect JMF4073 sensitivity.

Since DNA replication stress can also be regulated by the redox status, we then measured ROS, GSSG, and reduced GSH in these cells. The results showed that the total levels of ROS and GSSG were higher in T315I-Bcr-Abl-32D than those in WT-Bcr-Abl-32D cells (Fig. 3c, d). Despite these differences, the total amount of reduced glutathione (GSH) in WT-Bcr-Abl-32D cells was significantly lower than that in untransformed and T315I-Bcr-Abl-32D cells (Fig. 3e). We further compared the biosynthesis GSH synthesis by flux analysis in WT- and T315I-Bcr-Abl-32D cells incubated with (U)-$^{13}$C-glutamine. The analysis revealed a significant increase in the m + 5 GSH level in T315I-Bcr-Abl cells as compared to that in WT-Bcr-Abl cells (Fig. 3f), suggesting that upregulation of GSH biosynthesis might have a functionality in counteracting the increase in ROS. Overall, our observations imply the relevance of JMF4073 sensitivity to high replication stress associated with low dTTP/GSH levels in WT-Bcr-Abl-32D cells.

## ATF4 activation upregulates GSH and dTTP pools in T315I-Bcr-Abl-32D cells

Next, we want to address the questions of what causes the increases in dTTP and GSH biosynthesis in T315I-Bcr-Abl and whether blocking the upregulation of these two factors can increase JMF4073 sensitivity. Of note, the analysis by western blot revealed that the levels of enzymes involved in dTTP biosynthesis, like R1 and R2 subunits of ribonucleotide reductase (RNR), thymidylate synthase (TS), thymidylate kinase (TMPK), and thymidine kinase 1 (TK1), were similar in WT and T315I-Bcr-Abl cells (Supplementary Fig. 2). To comprehensively understand the gene expression changes responsible for the differences in dTTP and GSH biosynthesis, we performed RNA sequencing analysis of WT- and T315I-Bcr-Abl-32D cells. The gene set enrichment analysis (GSEA) of gene ontology (GO) pathways of RNA sequencing data revealed that T315I-Bcr-Abl cells have an increase in ATF4-mediated integrated stress response (ISR) signaling (Supplementary Fig. 3), which is known to upregulate the expression of genes involved in amino acid transport or synthesis, one-carbon metabolism, and glutathione biosynthesis[33,34] (Fig. 4a, b). RT-qPCR analyses validated the upregulation of ATF4 target genes, including *SLC1A1, SLC7A8*, phosphoserine aminotransferase 1 (*PSAT1*), and serine hydroxymethyltransferase 2 (*SHMT2*) in T315I-Bcr-Abl cells (Fig. 4c). Meanwhile, the expression levels of a number of GSH transferase genes were downregulated in T315I-Bcr-Abl-32D cells. The Western blot analysis confirmed that T315I-Bcr-Abl-32D cells clearly expressed higher level of ATF4 (Fig. 4d). Notably, SHMT2 is a mitochondrial enzyme contributory to one-carbon metabolism by converting serine and tetrahydrofolate (THF) into glycine for GSH synthesis and a one-carbon unit, 5,10-methylenetetrahydrofolate (5,10-CH$_2$-THF), for dTTP synthesis[35], respectively. Knockdown of ATF4 by lentiviral shRNA infection reduced RNA levels of *PSAT1* and *SHMT2* in T315I-Bcr-Abl-32D cells (Fig. 4e, f). Knockdown of *SHMT2* in T315-Bcr-Abl-32D cells led to about 50% reduction in dTTP pool in these cells (Fig. 4g). Therefore, ATF4-mediated upregulation of SHMT2 contributes to a higher level of dTTP in T315I-Bcr-Abl-32D cells.

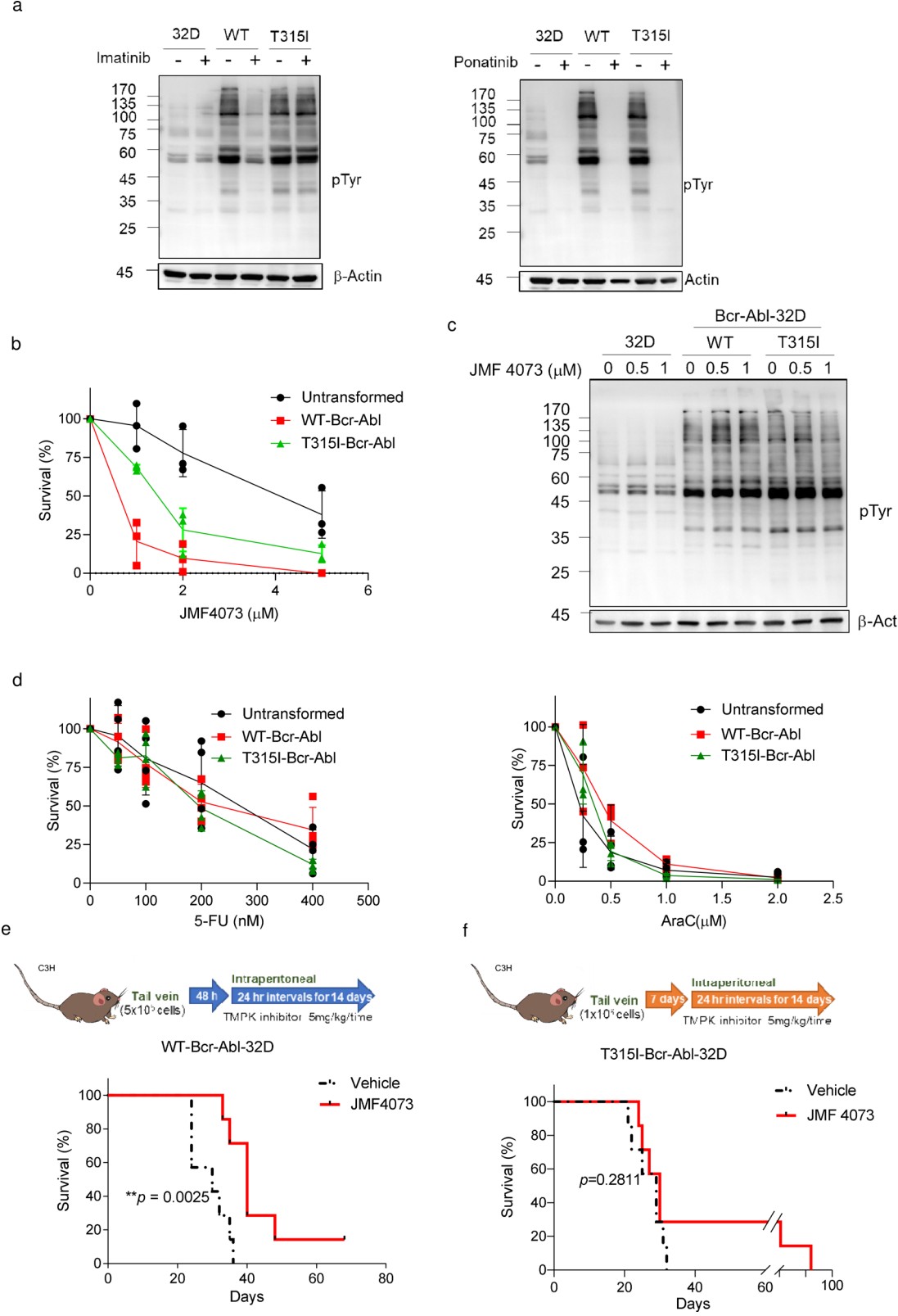

The ATF4-mediated upregulation of amino acid transporters and one-carbon metabolism, including *SLC1A1, SLC7A8*, and *SHMT2*, presumably increases the supply of glutamate, cysteine, and glycine, which might act in concert for the biosynthesis of GSH. We then tested the effect of ATF4 knockdown on GSH level in T315I-Bcr-Abl-32D cells. In agreement, ATF4 knockdown significantly decreased GSH level in these cells (Fig. 4h). Moreover, ATF4 knockdown in T315I-Bcr-Abl cells also slowed down DNA replication fork progression (Fig. 4i). Thus, the increase of ATF4-mediated integrated stress response (ISR) in T315I-Bcr-Abl 32D cells remodels dTTP and GSH biosynthesis, thus avoiding DNA replication stress.

**Fig. 2 | JMF4073 eliminates WT-Bcr-Abl-transformed myeloid cells in vitro and in vivo. a** The comparison of overall pTyr in unstranformed, WT, and T315I Bcr-Abl -transformed 32D myeloid progenitor cells, After the treatment of imatinib (2 µM) and ponatinib (2 µM) for 4 h, cell lysates were analyzed by Western blot using phosphotyrosine (pY99) antibody. Cells were treated with JMF4073 at the indicated concentrations for (**b**) viability assays and (**c**) Western blot of tyrosine phosphorylation. **d** Cells were treated with 5-FU, and ara-C at the indicated concentrations for viability assays. Data are represented as means ± S.D., n = 3 biological replicates. **e** C3H/HeNCrNarl mice were intravenously injected with $5 \times 10^5$ cells of WT-Bcr-Abl-32D cell. After 48 h of transplantation, mice were treated with vehicle (n = 7), or JMF4073 (5 mg/kg/time, n = 7) by intraperitoneal injection at 24 h intervals for 14 days. **f** C3H/HeNCrNarl mice were intravenously injected with $1 \times 10^6$ cells of T315I-Bcr-Abl-32D cell. After 7 days of transplantation, mice were treated with vehicle (n = 7), or JMF4073 (5 mg/kg/time, n = 7) by intraperitoneal injection at 24 h intervals for 14 days. The Kaplan–Meier plot shows the survival of mice with treatment as indicated. Asterisks denote *p < 0.05, **p < 0.01, ***p < 0.001, or ****p < 0.0001 from unpaired two-tailed Student's t-test or Log-rank (Mantel–Cox) test. The mice images are hand-drawn using the free Samsung PENUP software.

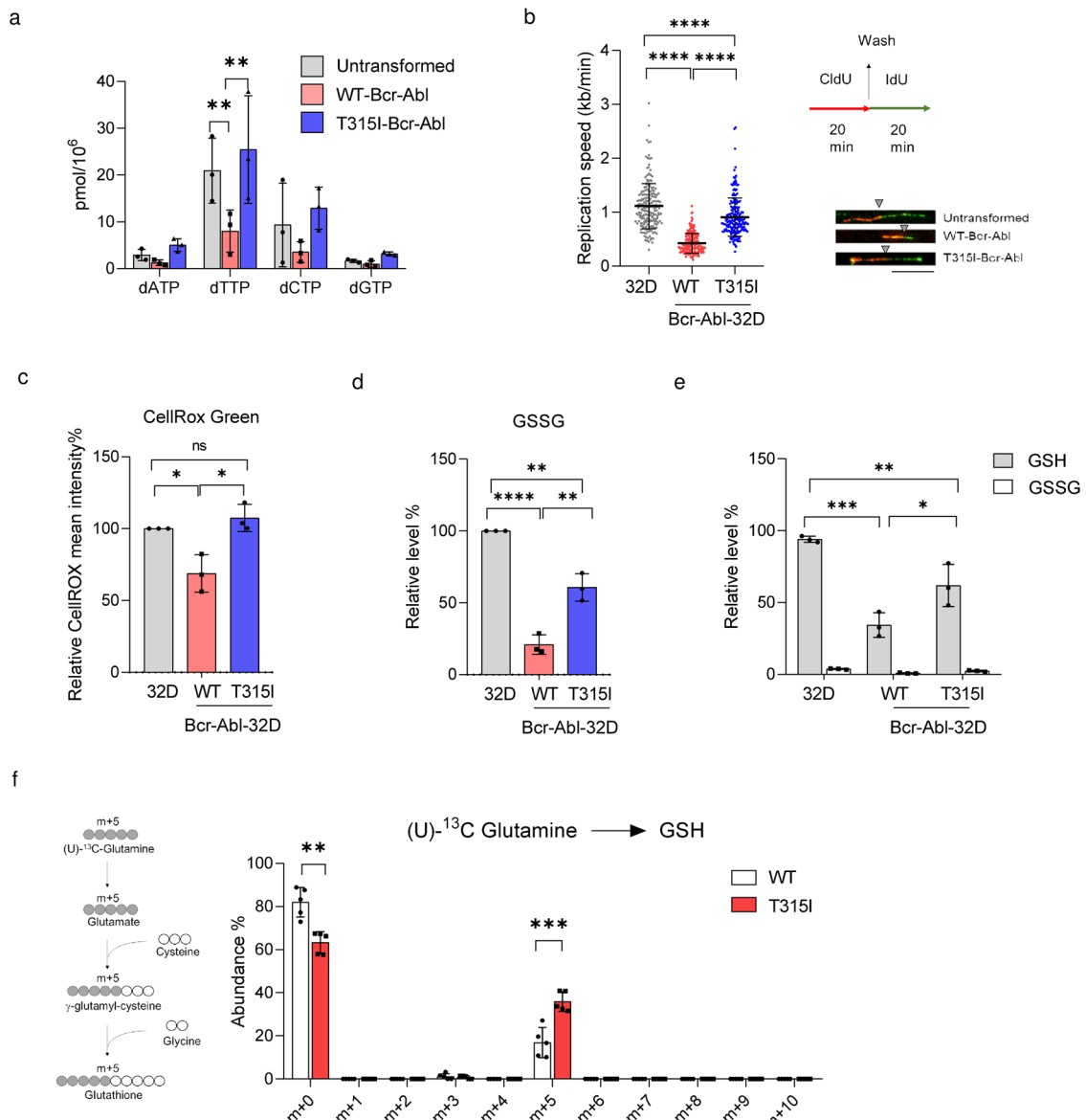

**Fig. 3 | The association of JMF4073 sensitivity with replication stress, low dTTP, and GSH pools.** Untransformed, WT-, and T315I-Bcr-Abl transformed 32D cells were subjected to (**a**) the measurement of four dNTP levels, **b** DNA fiber analysis to determine the replication fork speed in DNA replication speed (n = 200 fibers for each cell line) (*Left*). The workflow of DNA fiber labeling (*Right*). The representative labeled fibers are shown below. Replication speed in kb/min was calculated by the measured length of IdU (green) linking to CldU (red) in µm with a conversion factor of 0.34 µm/kb divided by the duration of the labeling pulse (Scale bar = 10 µm). The gray triangle indicates the boundary between red fluorescence and green fluorescence. **c** The measurement of intracellular ROS by the fluorescence intensity of CellROX Green staining using flow cytometric analysis. Data are presented as mean intensity relative to untransformed 32D cells. **d**, **e** The measurement of GSSG and GSH. Data are presented relative to untransformed 32D cells. **f** WT- and T315I-Bcr-Abl-32D cells after incubation with 4 mM U-$^{13}$C glutamine medium for 2 h. Schematic illustration of the metabolic path U-$^{13}$C glutamine incorporation into the GSH (*left*). Gray and white circles indicate $^{13}$C-label, and unlabeled $^{12}$C, respectively. The traced isotopologue abundance of GSH is shown (*right*). Data are represented as means ± S.D., n = 3 biological replicates. Asterisks denote *p < 0.05, **p < 0.01, ***p < 0.001, or ****p < 0.0001 from unpaired two-tailed Student's t-test.

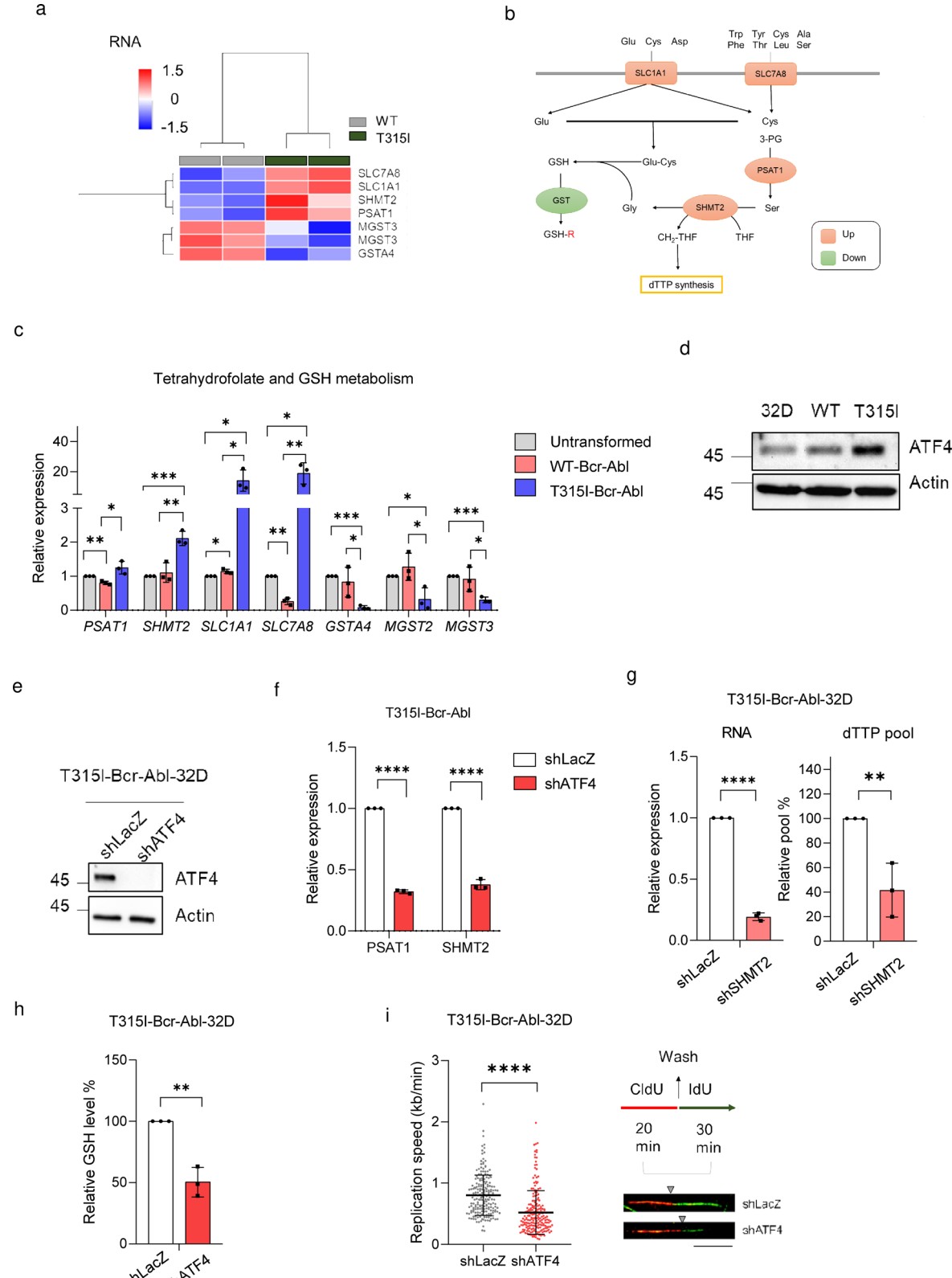

**Glutathione but not dTTP level determines JMF4073 susceptibility**

Next, we investigated whether the decrease in dTTP pool by *SHMT2* knockdown could induce DNA replication stress and increase susceptibility to JMF4073 in T315I-Bcr-Abl-32D cells. Intriguingly, the decrease in dTTP pools through *SHMT2* knockdown was insufficient to change the speed of

DNA replication fork progression and JMF4073 sensitivity in T315I-Bcr-Abl-32D cells (Fig. 5a, b), suggesting that dTTP pools might not be the critical factor for JMF4073 susceptibility.

We then tested whether the high GSH level is responsible for less susceptibility to JMF4073 in T315I-Bcr-Abl-32D cells. To this end, buthionine sulphoximine (BSO), which inhibits γ-glutamylcysteine synthetase[36],

**Fig. 4 | Upregulation of ATF4-mediated network increases dTTP pool and GSH biosynthesis in T315I-Bcr-Abl 32D cells. a** The heat map of RNA-sequencing data for genes significantly altered in dTTP synthesis and glutathione metabolism, as determined by Ingenuity Pathway Analysis (IPA), sorted based on log2 fold changes >1 or < -1 with adjusted p < 0.05. **b** A schematic diagram illustrates genes linked to dTTP synthesis and GSH metabolism. Genes upregulated in T315I-Bcr-Abl-32D cells are shown in orange and downregulated in green. **c** The RT-qPCR analysis of genes indicated in (**a**). Data were normalized using GAPDH, and the expression levels were expressed relative to those in untransformed 32D cells. **d** Immunoblotting of ATF4 in untransformed, WT-, and T315I-Bcr-Abl

transformed 32D cells. T315I-Bcr-Abl-32D cells infected with shLacZ and shATF4 lentivirus for 48 h were subjected to (**e**) immunoblotting and (**f**) RT-qPCR analysis to assess *PSAT1* and *SHMT2* levels. **g** RT-qPCR analysis of *SHMT2* and the quantitation of dTTP level in T315I-Bcr-Abl-32D cells after infection with shLacZ and shSHMT2 lentivirus. **h, i** T315I-Bcr-Abl-32D cells after infection with shLacZ and shATF4 lentivirus for 48 h were subjected to (**h**) GSH level and (**i**) the DNA fiber analysis (n = 200, scale bar = 10 μm). The gray triangle indicates the boundary between red fluorescence and green fluorescence. Data are presented as means ± S.D, from three independent experiments. Asterisks denote **p < 0.01, ***p < 0.001, or ****p < 0.0001 from unpaired two-tailed Student's t-test.

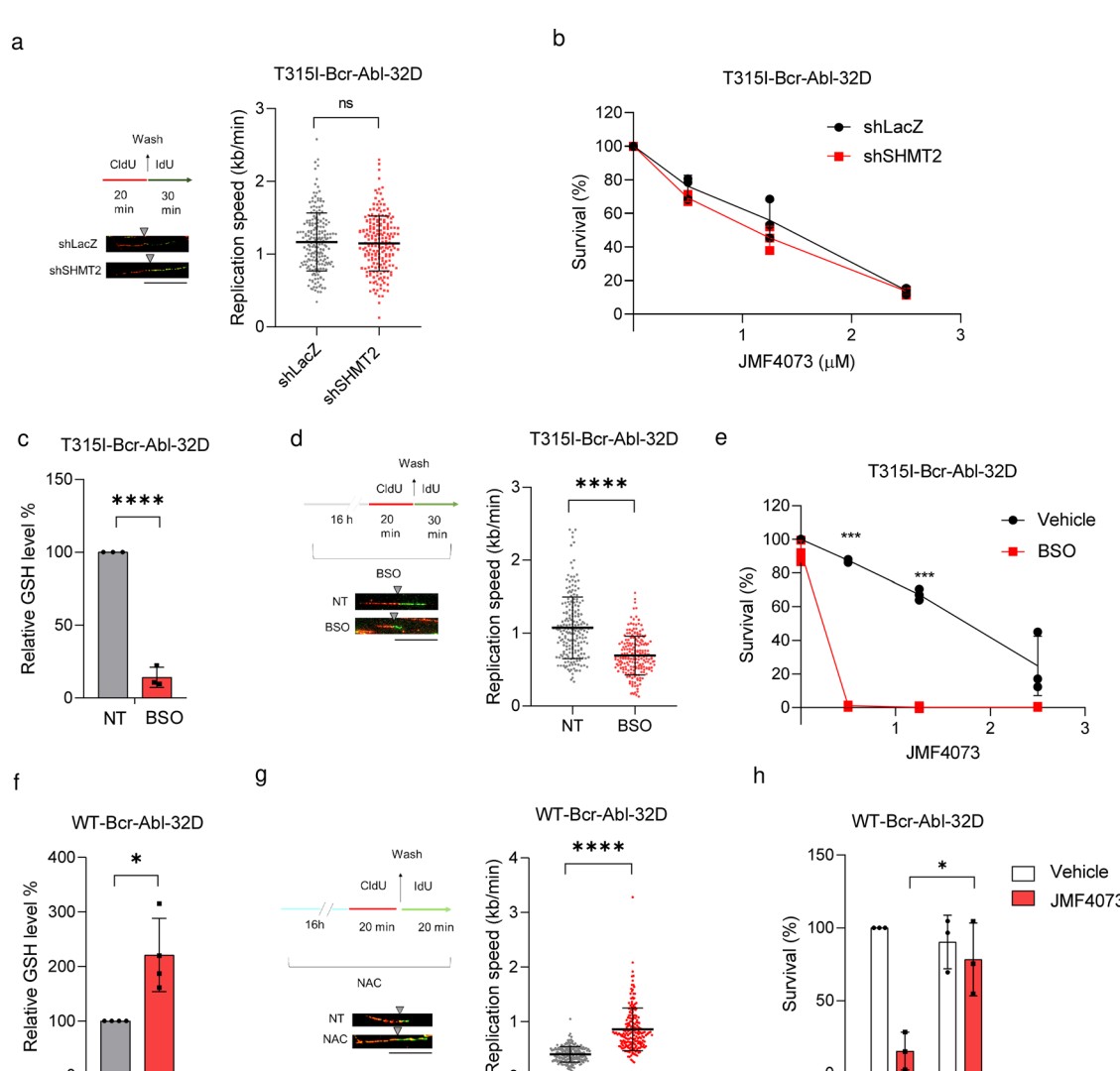

**Fig. 5 | GSH biosynthesis but not dTTP determine JMF4073 susceptibility.** T315I-Bcr-Abl-32D cells after infection with shLacZ and shSHMT2 lentivirus were examined for (**a**) DNA fiber analysis as described in the legend to Fig. 3b (scale bar = 10 μm), and (**b**) JMF4073 sensitivity analysis by viability assays. T315I-Bcr-Abl-32D cells after treatment with BSO (12.5 μM) for 16 h were subjected to (**c**) the measurement of GSH level, (**d**) DNA fiber analysis, and (**e**) JMF4073 sensitivity.

WT-Bcr-Abl-32D cells after treatment with N-acetyl cysteine (NAC, 1 mM) for 16 h for (**f**) the measurement of GSH level, (**g**) DNA fiber analysis, and (**h**) JMF4073 sensitivity. The viability of cells treated with NAC in combination with JMF4073 (0.5 μM) for 72 h was determined. Data are presented as means ± S.D., n = 3 biological replicates. Asterisks denote *p < 0.05, **p < 0.01, or ****p < 0.0001 from unpaired two-tailed Student's t-test.

was used to treat T315I-Bcr-Abl-32D cells. As expected, BSO treatment reduced the cellular level of GSH together with the lengths of replication tracks (Fig. 5c, d). Moreover, BSO co-treatment markedly increased JMF4073 sensitivity (Fig. 5e). Since BSO treatment alone did not suppress the growth of T315I-Bcr-Abl cells, a synergistic effect of JMF4073 in combination with BSO on elimination of these cells was very clear. Conversely,

WT-Bcr-Abl cells after treatment with N-Acetyl-Cysteine (NAC), a GSH precursor that increased the cellular GSH level, exhibited not only increased lengths of replication tracks but also insensitive to JMF4073 (Fig. 5f–h). Altogether, these data suggest that the upregulation of GSH, rather than the dTTP pool, is responsible for avoiding oncogene-induced replication stress and less susceptibility to JMF4073 in T315I-Bcr-Abl-32D cells.

### Blocking GSH synthesis in combination with JMF4073 depletes dTTP/dCTP pools to eradicate T315I-Bcr-Abl leukemia

We hypothesize that reducing the GSH level to increase replication stress can render T315I-Bcr-Abl leukemia more susceptible to JMF4073 in vivo. As mentioned, BSO treatment was able to synergize the effect of JMF4073 on eliminating T315I-Bcr-Abl cells. However, we found that BSO in combination with JMF4073 also severely suppressed the growth of 32D myeloid progenitor cells (Supplementary Fig. 4), suggesting the lack of selectivity for leukemia cells. It has been reported that treatment with UK-5099, an inhibitor of mitochondrial pyruvate carrier (MPC), reduces the cellular GSH level by diverting glutamine metabolism from GSH synthesis in tumor cells[37]. In agreement, UK-5099 treatment was able to reduce GSH levels and slow replication fork progression in T315I-Bcr-Abl cells. It was noted that the effects of UK-5099 treatment on GSH reduction and replication stress were less in 32D cells (Fig. 6a, b). As a consequence, UK-5099 treatment sensitized T315I-Bcr-Abl-32D to JMF4073 but not 32D progenitor cells (Fig. 6c). We further analyzed dTTP and dCTP pool after the combinatory treatment of UK-5099 with JMF4073 and found the exhaustion of dTTP and dCTP pools in T315I-Bcr-Abl-32D cells (Fig. 6d). We then performed in vivo therapy of UK-5099 with JMF4073. After transplanting T315I-Bcr-Abl-32D/EGFP cells into C3H/HeNCrNarl mice for 7 days, the mice were treated with daily intraperitoneal injections of UK-5099 alone or in combination with JMF4073 for 14 days (Fig. 6e, f). The survival of T315I leukemia mice was not affected by UK-5099 therapy. As a contrast, the combination of UK-5099 with JMF4073 therapy prolonged the survival. Moreover, mice blood analysis revealed marked decreases in the population of T315I-Bcr-Abl-32D/EGFP + cells by the combination of JMF4073 with UK-5099 treatment (Fig. 6g), along with the spleen size (Fig. 6h).

### Synthetic lethality by targeting glutathione and pyrimidylate kinases in human CML-BC cell lines

Finally, we further used several human blast-crisis CML cell lines, including K562, KOPM28, and TCCS, to verify the relationship between ATF4, GSH, and replication stress. The results showed the correlation of ATF4 with GSH level and the replication fork length. Among these human cell lines, the levels of ATF4 and GSH were lowest in K562 cells, while much higher in TCCS and KOPM28 cells (Fig. 7a–c). In agreement, JMF4073 sensitivity was higher in K562, less in TCCS, and insensitive in KOPM28 (Fig. 7d). Thus, the JMF4073 sensitivity is inversely correlated with ATF4 and GSH levels. For K562 cells, UK-5099 treatment was sufficient to bring down the cellular level of GSH. As for KOPM28 and TCCS cells, GSH level remained unchanged after UK-5099 treatment (Fig. 7e). We then treated these cells with a low dosage of erastin, the inhibitor of cysteine transporter xCT[38], which drastically reduced GSH level and induced replication stress (Fig.7f, g). We then tested whether the treatment with erastin at 0.5 μM, a sub-lethal dosage, could sensitize KOPM28 and TCCS CML-BC lines to JMF4073. The results showed that erastin markedly increased the JMF4073 sensitivity in these two CML-BC cell lines (Fig. 7h). Thus, synthetic lethality is achieved by co-targeting CMPK/TMPK and GSH in CML-BC cells that have high level of GSH. On the other hand, these data also imply that JMF4073 co-treatment enlarges the therapeutic window of erastin.

## Discussion

In this study, we developed a small compound, termed JMF4073, which inhibits CMPK and TMPK. WT- and T315I-Bcr-Abl-32D cells show differential vulnerability to JMF4073 treatment. In order to substantiate the therapeutic potential of JMF4073 for TKI-resistant T315I-Bcr-Abl cells, we investigated the mechanism responsible for the differential response to JMF4073. Our data analyses showed the deficiencies in GSH and dTTP with replication stress in WT-Bcr-Abl-32D cells as a contrast to high GSH and dTTP pools without replication stress in T315I-Bcr-Abl-32D cells. By analyzing RNA sequencing data, we found the increase in ATF4-mediated transcriptional network for the upregulation of *SHMT2* for dTTP in T315I-Bcr-Abl-32D cells. This explained why dTTP pool is particularly higher in

these cells. The increase of ATF4 in T315I-Bcr-Abl cells is also responsible for more GSH biosynthesis. Likely, T315I-Bcr-Abl cells develop antioxidative pathways and metabolic rewiring via ATF4-mediated transcriptional network for growth fitness. Blocking GSH synthesis by UK-5099 or BSO converts T315I-Bcr-Abl cells highly sensitive to JMF4073. As a contrast, knockdown of *SHMT2* to reduce dTTP levels had no effect on JMF4073 in these cells. Since co-treatment of UK-5099 and JMF4073 depleted dTTP pool and suppressed the growth of T315-Bcr-Abl-32D cells, we proposed that reducing GSH biosynthesis induces not only oxidative stress but also replication stress in tumor cells, which demands the synthesis of dTTP and dCTP for survival. Thus, GSH is the factor that determines JMF4073 sensitivity.

During the replication fork progression, TIMELESS forms a complex with TIPIN in the replisome to facilitate the progression of DNA replication fork[17,39,40]. This process has been demonstrated to involve the association with oligomerized PRDX2[17]. Elevated ROS increases PRDX2 monomer formation, which causes TIMELESS-TIPIN dissociation from the replisome and replication slowdown, thereby inducing replication stress. The study has shown that PRDX2 is not required for TIMELESS-TIPIN-regulated replication fork progression but plays a redox sensor to control this process. The oligomerization of PRDX2 is inhibited by ROS-induced hyperoxidation to sulfinic acid at cysteine residue. Since PRDX2 has been shown to be glutathionylated by GSH at its cysteine residues to protect against its hyperperoxidation[41], it is possible that elevated GSH levels in T315I-Bcr-Abl cells might prevent PRDX2 monomer formation, therefore abrogating ROS-induced replication stress. However, this possibility remains to be investigated.

This study found higher levels of ATF4 not only in T315I-Bcr-Abl-32D cells but also in human CML blast-crisis cell lines. Consistent with the ATF4-mediated transcriptional control for GSH biosynthesis, these cells all showed higher levels of GSH and the lack of replication stress. So, the question is why the level of ATF4 is increased in these cell lines. It is well established that cap-independent translation of ATF4 requires eIF2α, which is activated by endoplasmic reticulum (ER), oxidative, or mitochondrial stress[42–44]. Through ATF4-mediated integrated stress responses, cancer cells acquire not only drug persistence, ferroptosis sensitivity, and also metastatic phenotype[45–47]. Our findings raise the question of whether ATF4-mediated ISR is generally present in these TKI-insensitive CML cells and what causes the activation of ATF4 translation in these cells.

Our data showed that JMF4073 sensitivity was inversely correlated with the cellular levels of GSH. Among three human CML blast-crisis lines, K562 cells are most sensitive to JMF4073. Another two CML lines have much higher levels of GSH and are insensitive to the combinatory treatment of JMF4073 and UK-5099, because UK-5099 treatment was unable to reduce GSH and induce replication stress. However, treatment with a low dosage of erastin was capable of drastically reducing GSH with replication stress induction. As a result, the combination of erastin and JMF4073 both at sub-micromolar concentration acts synergistically to cause lethality in these blast-crisis CML cells. Very obviously, the combination with JMF4073 indeed enlarges the therapeutic window of erastin. Thus, GSH and ATF4 levels in CML cells are the markers to indicate JMF4073 sensitivity and determine the selection of GSH blocker for the combination with JMF4073 in therapy. Our data suggest a potential therapeutic approach involving co-targeting GSH and dTTP pools. Although ponatinib is an FDA-approved TKI that directly targets T315I-Bcr-Abl leukemia, its association with cardiovascular toxicity is a significant concern in clinical therapy. Therefore, the combination of JMF4073 with GSH blocker is an alternative option. However, the limitation of this combinatory therapy remains the choice of GSH blocker that might also elicit toxicity.

## Methods
### Cell lines and cell culture
Murine WEHI, 32D (32Dcl3), WT, and T315I-BcrAbl-transformed 32D cells were generously provided by Dr. Robert Arlinghaus (The University of

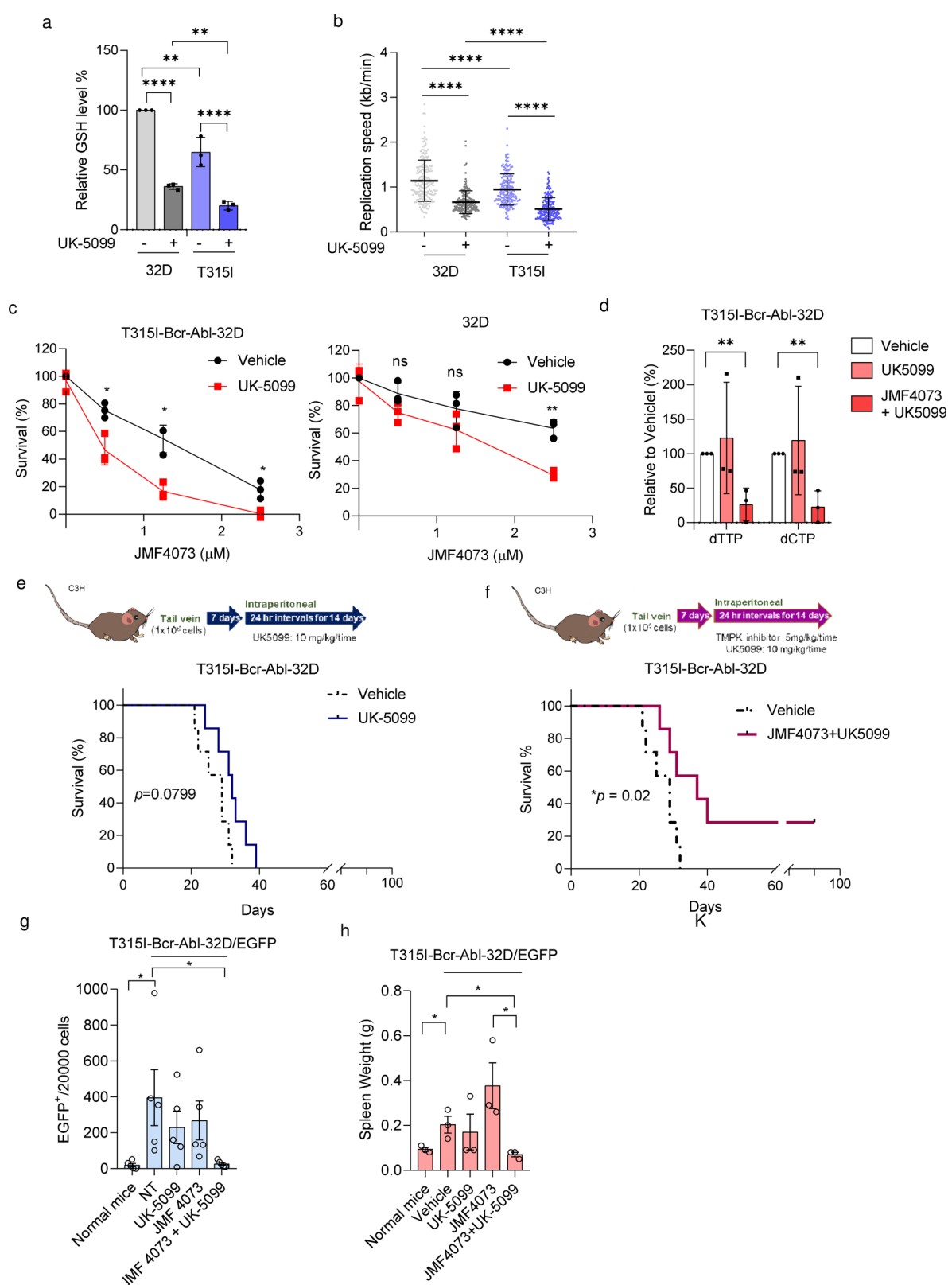

Texas MD Anderson Cancer Center, Houston, TX, USA.). K562 cells were obtained from BCRC (Bioresource Collection and Research Center, Taiwan), and TCCS/KOPM28 from Takeshi Inukai (Department of Pediatrics, School of Medicine, University of Yamanashi, Chuo, Japan). HEK-293T were from American Type Culture Collection (ATCC), and cultured in Dulbecco's modified Eagle's medium (DMEM) (Cytiva, SH30003.02) supplemented with 10% fetal bovine serum (FBS, Cytiva). WEHI, 32D (32Dcl3), WT, T315I-BcrAbl-transformed 32D, and TCCS cells were maintained in RPMI 1640 medium (Sigma) supplemented with 10% heat-inactivated FBS, 1% Streptomycin/Penicillin, 1 mM sodium pyruvate, and

**Fig. 6 | The combinatory treatment of JMF4073 and UK-5099 depletes dTTP/dCTP pools and eliminates T315I Bcr-Abl-32D cells in vivo.** T315I-Bcr-Abl-transformed and untransformed 32D cells were treated with UK-5099 (25 μM) for 16 h. These cells were subjected to (**a**) GSH measurement, and (**b**) DNA fiber assay. **c** JMF4073 sensitivity assay. **d** dTTP and dCTP pools measurement of T315I-Bcr-Abl-32D cells after treatment with vehicle, UK-5099 alone, and the combination of UK-5099 and JMF4073. **e, f** C3H/HeNCrNarl mice were intravenously injected with $1 \times 10^{6}$ cells of T315I-Bcr-Abl-32D cell. After 7 days of transplantation, mice were treated vehicle (n = 7), UK-5099 (10 mg/kg/time, n = 7) (**e**) or JMF4073 (5 mg/kg/time) + UK-5099 (10 mg/kg/time) (n = 7) (**f**) by intraperitoneal injection at 24 h

interval for 14 days. The Kaplan–Meier plot shows the survival of mice with treatment as indicated. **g** flow cytometry analysis of peripheral blood (n = 5) and (**h**) Spleen weights (n = 3) from T315I-Bcr-Abl-32D/EGFP+ bearing mice treated with vehicle, UK-5099, JMF4073, or JMF4073 combined with UK-5099. The number of EGFP+ cells in mice with T315I-Bcr-Abl-induced CML was determined on day 30 after transplantation. Data are represented as means ± S.D., n > 3 biological replicates. Asterisks denote *p < 0.05 from unpaired two-tailed Student's t-test or Log-rank (Mantel–Cox) (**f, g**) test. The mice images are hand-drawn using the free Samsung PENUP software.

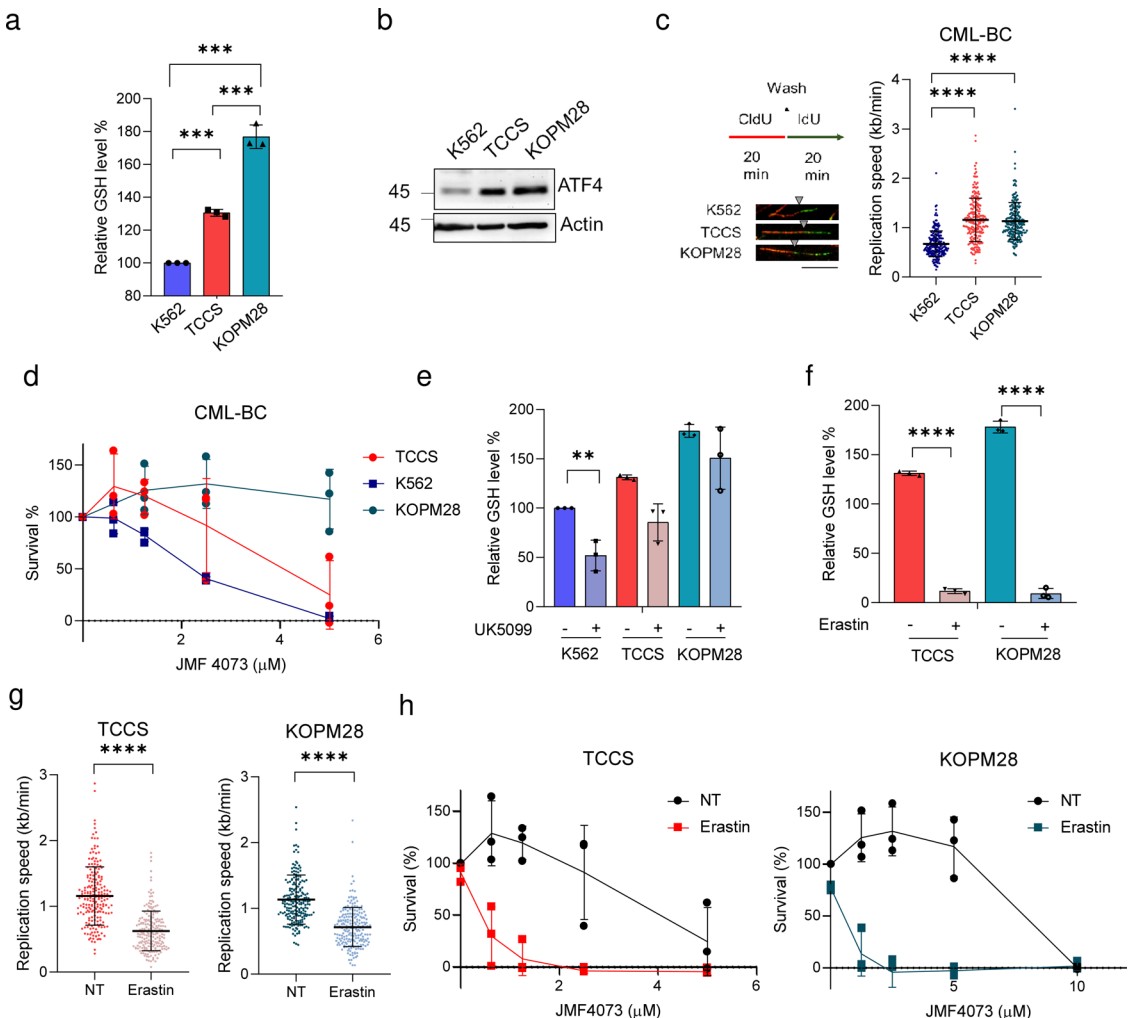

**Fig. 7 | Synthetic lethality by GSH blocker and JMF4073 in human CML-BC cells.** K562, TCCS, and KOPM28 were subjected to (**a**) GSH measurement, (**b**) immunoblotting of ATF4, and (**c**) DNA fiber analysis. **d** K562, TCCS and KOPM28 cells were treated with JMF4073. After 3 days, cells were analyzed by viability assays. GSH level in K562, TCCS, and KOPM28 cells after treated with (**e**) UK-5099 (50 μM) or (**f**) erastin (0.5 μM) for 16 h for GSH measurement. **g** After treatment with erastin

(0.5 μM) for 16 h, DNA fiber analyses were performed in TCCS (n = 200) and KOPM28 cells (n = 200). **h** JMF4073 sensitivity in TCCS and KOPM28 cells after pretreatment with erastin (0.5 μM) for 16 h. All data are presented as means ± S.D. from 3 independent experiments. Asterisks denote **p < 0.01, or ****p < 0.0001 from unpaired two-tailed Student's t test.

10 mM HEPES. The 32D cell medium included 10% conditioned medium from WEHI-3B cell culture as interleukin-3 (IL-3) source. K562 and KOPM28 cells were cultured in Iscove's Modified Dulbecco's Medium (IMDM, Sigma, I6529) with 10% FBS.

### Compounds

Imatinib (HY-15463) and Ponatinib (HY-12047) were from MedChem-Express(MCE); UK-5099 (PZ0610, Sigma), Erastin (E7781), N-acetyl-cysteine (A7250), GSH-MEE (353905), NADH (N8129), dTMP (T7004), CMP (C1131), and ATP (A-6419) were purchased from Sigma.

### Chemistry

Compound JMF4073, Tert-butyl 4-(2-(5-fluoro-3-oxobenzo[d]isothiazol-2(3H)-yl) acetyl) piperazine-1-carboxylate, was synthesized using 2,5-difluorobenzonitrile as the starting materials. See supplementary methods for synthetic schemes and procedures with NMR spectrums.

### Purification of recombinant proteins

GST-hTMPK and His-tagged CMPK were purified from pGEX-2T-hTMPK and pET28c-CMPK–transformed BL21(DE3) E. coli, respectively, as described previously[20,48].

## TMPK and CMPK activity assay by NADH-coupled enzymatic assay

Activity assay was measured using the spectrophotometric method by coupling ADP formation to NADH oxidation catalyzed by pyruvate kinase and lactate dehydrogenase as described previously[20]. Each reaction contained 0.4 μg TMPK or 0.3 μg CMPK was incubated at 25 °C followed by using Tecan Spark 10 microplate reader to detect the change in NADH measurement at 340 nm absorption according to the manufacturer's instructions.

## Cell viability assay

Cells were seeded at the density of $1 \times 10^3$ cells/well in 96-well microplate. After 3 days, cell viability was determined using the CCK-8 cell viability. The viability was measured by using Tecan Spark 10 microplate reader at 450 nm according to the manufacturer's instructions.

## Reactive oxygen species detection

Cells were processed as per manufacturer's (Invitrogen™, C10444) instructions. Cells with ROS were analyzed and quantified by flow cytometry (FACS Calibri).

## Mouse CML induction and in vivo therapy

The animal studies were approved by the biosafety committee at National Taiwan University and conformed to the national guidelines and regulations (IACUC # 20201063). Female C3H/HeNCrNarl mice were used at 6–8 weeks of age (National Laboratory Animal Center, Taiwan). Additional details are in Supplementary Methods.

## Viral infection

For lentiviral production, HEK293T cells were co-transfected with 4 μg Δ8.91, 1 μg pVSVG, 5 μg of lentivector (shRNA targeting *SHMT2*: CCA-GAATTTGTAGCTGAGATT, and *ATF4*: GCGAGTGTAAGGAGCTA-GAAA, obtained from RNA Technology platform and Gene manipulating core, Academia Sinica) by using 30 μL of Turbofect (Thermo, R0533) in 1 mL of Opti-MEM (Gibco, 31985070) according to the manufacturer's instructions. Viral supernatants were collected 48 h and used to infect T315I-Bcr-Abl-32D cells in the presence of 8 μg/ml polybrene. Cells were then selected and maintained in culture media containing 1 μg/mL of puromycin.

**Antibodies**. Specific primary antibodies: Akt (2532 S, 1:2000), pS473-Akt (9271 S, 1:1000), S6K (9202 S, 1:2000), pT389-S6K (9205 S, 1:1000), Stat5 (9420 S, 1:10000), and pY694-Stat5 (9356 S, 1:1000) were obtained from Cell Signaling. Anti-phospho-Tyr99 (sc-7020, 1:1000) from Santa Cruz. β-Actin GeneTex (GTX109639, 1:1000); RRM1 (sc-11733, 1:1000), and RRM2 (sc-10846, 1:1000) from Santa Cruz. anti-TMPK and anti-CMPK were prepared as described previously[49].

## GSH and GSSG measurement

The GSH/GSSG-Glo™ Assay Kit (Promega, V6611) was used for luminescent-based measurement of cellular GSH and GSSG by Tecan Spark 10 microplate reader.

## Metabolic flux by U-¹³C-glutamine tracing measurement

Cells in 10-cm dishes were washed twice for metabolite tracing by glutamine-free RPMI-1640 medium. For glutamine tracing, 4 mM of U-$^{13}$C-glutamine was added to the glutamine-free RPMI-1640 medium with 10% v/v HI-FBS, 1 mM HEPES, and 1 mM sodium pyruvate. After 2 h incubation, cells were washed twice with cold PBS and were extracted with 80% ice-cold methanol at $2 \times 10^6$ cells/mL. Samples were incubated at −80 °C for overnight. After centrifugation, supernatants were transferred to fresh tubes, evaporated using a speed-vac, and subjected to mass spectrometry analysis. Additional details are in Supplementary Methods.

## Measurement of dNTP pools by RCA-qPCR assay

Cells ($10^6$) were extracted with 1 ml of ice-cold 80% methanol for dNTP pool measurement by RCA-qPCR assay as described previously[50]. Briefly, the methanol extracts were centrifuged and subjected to chloroform extraction. After phase separation, the upper phase was collected and vacuum-dried. The dried residues were dissolved in water, in which $1 \times 10^4$ cells/ 5 μL were used for the phi29 polymerase-based RCA reaction mixture. The RCA products were determined by the subsequent addition of another 10 μL of qPCR reaction mixture (PCR Biosystem). Additional details and oligonucleotides used in RCA-qPCR are listed in Supplementary Methods.

## DNA fiber assay

The DNA fiber assay was performed as described previously[51]. Briefly, cells were sequentially treated with the medium containing 25 μM of CldU (Sigma-Aldrich, C6891) and then 250 μM of IdU (Sigma-Aldrich, I7125) for 20 min. After washing, cells were suspended in PBS (1000 cells/ μL). 2 μL of cell suspension was spotted on slides. After 5 min, 10 μL spread solution (200 mM Tris-HCl, pH 7.5, 50 mM EDTA, and 0.5% SDS) was spotted on the cell droplet and spread on slides. The slides were then fixed with methanol/acetic acid (3:1), followed by denaturation with 2.5 M HCl. After blocking with 5% BSA in TBST, slides were stained with rat anti-BrdU antibody (which detects CldU but not IdU, OBT0030, 1:2000, AbDSerotec) and mouse anti-BrdU antibody (which detects IdU but not CldU, 7580, 1:1000, BD Biosciences) at 4 °C for overnight. The next day, slides were washed with TBST, followed by staining with the secondary antibodies of goat anti-rat IgG conjugated with TRITC (Jackson Lab) and goat anti-mouse IgG conjugated with FITC. Images of DNA fibers were acquired using a microscope (Olympus). The lengths of red- and green-labeled fibers were determined by FluoView3.0 software (Olympus).

## RNA extraction for RNA sequencing and RT-qPCR

Total RNA was extracted using TRIzol reagent and subjected to RNA sequencing (See Supplementary Method for detail). 1 μg RNA was reverse transcribed into cDNA using the Moloney Murine Leukemia Virus Reverse Transcriptase (M-MLV RT, Promega) following the manufacturer's instructions. qRT-PCR analyses were performed using 2x qPCRBIO Sybr Green blue Mix (PCR Biosystem) on a Real-Time PCR Detection System (Applied Biosystems). The primers used are listed in Supplementary Methods.

## RNA sequencing analysis

RNA (1 μg) per sample was used for sequencing libraries generation by using KAPA mRNA HyperPrep Kit (KAPA Biosystems, Roche, Basel, Switzerland) following the manufacturer's recommendations to remove ribosomal RNA. The libraries were then purified using KAPA Pure Beads system and assessed on the Qsep 100 DNA/RNA Analyzer (BiOptic Inc., Taiwan) for quality check. The PCR products were subjected to high-throughput sequencing (Illumina NovaSeq 6000 platform) with a paired-end 150 base pair strategy.

The raw reads were trimmed and aligned to the reference *Mus musculus* genome by the HISAT2 software. The quantification and normalization (DESeq method) and further downstream analyses of identification of differentially expressed genes (DEGs) were done by using R package. The resulting p-values were adjusted using the Benjamini and Hochberg's approach for controlling the FDR. Genes exhibiting false discovery rate (FDR)-adjusted $P < 0.05$ and the absolute fold changes (FC) > 2 were considered significant and subjected to further analysis. GO and KEGG pathway enrichment analysis of DEGs were conducted using clusterProfiler. Gene set enrichment analysis (GSEA) was performed with 1000 permutations to identify enriched biological functions and activated pathways from the molecular signatures database (MSigDB), which included hallmark gene sets, positional gene sets, curated gene sets, motif gene sets, computational gene sets, GO gene sets, oncogenic gene sets, and immunologic gene sets.

Primers used in RT-qPCR for RNA sequencing validation are listed in Supplementary methods.

## Statistics and reproducibility

Data are presented as the mean ± S.D. Statistical comparisons between means were performed using an unpaired Student's t-test to assess differences between two groups. The log-rank test was used for Kaplan–Meier analysis. All in vitro experiments were performed at least three times. A $p < 0.05$ was considered indicative of statistical significance.

## Reporting summary

Further information on research design is available in the Nature Portfolio Reporting Summary linked to this article.

## Data availability

All data associated with this study are present in this paper or the supplementary information. RNA sequencing data are deposited in the National Center for Biotechnology Information's Sequence Read Archive (PRJNA812502). All source data behind each graph are shown in Supplementary Data 1.

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

## Acknowledgements

We are grateful for the support from the Laboratory Animal Center at National Taiwan University, College of Medicine and Flow Cytometric Analyzing and Sorting Core of the First Core Laboratory, National Taiwan University College of Medicine for providing the service. We thank Robert Arlinghaus (The University of Texas MD Anderson Cancer Center, Houston, TX, USA) for providing untransformed, WT- and T315I-Bcr-Abl transformed 32D cell lines. We thank Takeshi Inukai (Department of Pediatrics, School of Medicine, University of Yamanashi, Chuo, Japan) for kindly providing KOPM28 and TCCS cell lines. We thank Hsueh-Tzu Shih and Chih-Wei Chen for the technical assistance. We thank Gong-Min Lin at Metabolomics Core Facility of Agricultural Biotechnology Research Center in Academia Sinica for the assistance of metabolomics analysis and [13]C metabolite tracing. We also thank Dr. Yu-Lun Kuo at BIOTOOLS Co., Ltd. in Taiwan for kindly supporting analysis RNA sequencing. This research is supported by grants from the National Science and Technology Council, Taiwan (NSTC 112-2326-B-002-006 and NSTC 112–2320-B-002-005).

## Author contributions

C.Y.H. performed protein purification, enzymatic kinetics, viability, co-factors, metabolic analysis, DNA fiber assay, animal experiments, data analysis, organization and paper writing. S.Y.W. and Y.H.C. performed animal experiments. H.Y.W. performed dNTP quantitation. C.Y.L. performed metabolomics analysis and [13]C metabolite tracing experiments. J.M.F. and T.J.Y. synthesized compounds. C.M.H. tested drugs susceptibility. Z.F.C. conceived, designed, and supervised this project and wrote the paper.

## Competing interests

The authors declare no competing interests.
