## [Peer Review File · Communications Biology]

Glutathione Determines Chronic Myeloid Leukemia Vulnerability to an Inhibitor of CMPK and TMPK

Corresponding Author: Professor Zee-Fen Chang

Version 0:

Reviewer comments:

Reviewer #1

(Remarks to the Author)

The manuscript reports the discovery and development of an inhibitor of thymidylate kinase (TMPK) and cytidylate kinase (CMPK), JMF4073, aimed as second line therapeutics to cure TKI resistance chronic myeloid leukemia (CML) caused by T315I Bcr-Abl mutation. They found the drug resistance of the first line therapeutics comes from upregulation of GSH synthesis. However, JMF4073 is potent with wt-Bcr-Abl cells, but not T315I-Bcr-Abl cells they aimed at. Therefore, they proposed a combination therapy of JMF4073 with GSH synthesis inhibitor to treat T315I resistance CML. This work provides insight of T315I resistance CML and the compound JMF4073 is a potent TMPK and CMPK inhibitor. However, the project is not well designed and the manuscript is not well organized. It needs major revision for publication.

Major points,

1. The project design and logical in the study is not clear. Basically, the manuscript is difficulty to read and understand. They need to have a clear description of their goals and logical at the beginning of the manuscript, such as in the Abstract and Introduction.
2. The title is confusing to start with. The relationship of Glutathione, Bcr-Abl-induced Replication Stress and Vulnerability, and Inhibitor of Pyrimidylate Kinases are confusing or misleading.
3. Abstract, it is not clear that this work is aimed at a metabolic study to understand T315I resistance CML or a drug discovery. Need to revise with a better logical flow.
4. Abstract. There are many confusions in the wording. Here is an example, "...T315I mutation to acquire resistance to non-toxic tyrosine kinase inhibitors (TKI)" – how about toxic TKI? All TKI are non-toxic or only resistance to non-toxic TKI?
5. Introduction. Need to provide more backgrounds for general readers and use general terms as much as possible. For example, a brief description of t(9;22) would be helpful. Some parts of the Results and Discussion should be moved to introduction to prepare readers early to understand the work. For example, some the first paragraph of Discussion provided the Aims of this work and should be moved to Introduction. Some concepts popped up in the Results such as "Timeless/Tipin with replisome" (line 97), "Slc1a1 and Slc7a" (line 108) should be briefly described in Introduction.
6. Introduction needs to be reorganized. Brief description of the results should be put at the end of the introduction, instead of in every paragraph.
7. In the results, for up or down regulation, should indicate clearly what it refers to and compares with. For example, line 135-137, Fig3c, PSAT and SHMT2 level in 32D also should be shown to compare.
8. Results. "Altogether, these data suggest that upregulation of GSH rather than dTTP pool is responsible for avoiding oncogene-induced replication stress in T315I-Bcr-Abl-32D cells." (line 145) – it is not very clear why draws this conclusion. Need more explanations.
9. Results. Some statements lack of reference. For example, (line 77) Compelling evidence has suggested that oncogene-induced replication stress is a driving force in cancer progression. – need references.
10. Discussion. Some interesting questions could be discussed here. For example, why only dTTP and dCTP pools were affected, but not purine pools? (Figure 2&4 can be used for answers). Also, limitation of the work should be discussed. For example, JMF4073 is failed as an T315I resistance CML therapeutic. It works only together with GSH inhibitors. At least three of GSH inhibitors are tried in this work, from BSO to UK-5099 to erastin, with increasing potency, which surely will lead to increased toxicity. As a prove of concept, this going-around approach really works? Or should think about designing a better inhibitor directly work at T315I BCR-ABL?

Minor points,

1. Many self-invented terms were used, which made the manuscript hard to read. For example, what is "dNTPs blockers" (line 28) and "nucleotide blockers" (line 63)? Do they mean compounds with nucleotide-like structure? If so, please define these terms.
2. Pyrimidylate Kinase is not a general term. Please make definition.
3. Need to work on English for the proper descriptions and words.

Reviewer #2

(Remarks to the Author)

The authors explored therapeutic application of targeting GSH metabolism in treating chronic myeloid leukemia cells upon inhibiting pyrimidylate 30 kinases (TMPK and CMPK) via a homemade drug JMF4073. Overall, the study presented an interesting findings that CML cells containing low levels of glutathione require dNTPs supply to overcome DNA damage and replication stress for survival. Blocking glutathione in combination with JMF4073 led to synthetic lethality in WT-Bcr-Abl-32D. There are only several questions needed to be addressed before publication.

- 1, How exactly GSH level regulated replication stress in leukemia cells? High GSH and NAC suppressed replication stress. However, GSH and NAC are wellknown antioxidants. So, the role of ROS should also be examined in the context of study.
- 2, The paper showed that wt Bcr-Abl might regulated enzymes invovled in the biosynthesis of dNTPs, such as SHMT2. What's the potential mechanisms?

Author Rebuttal letter:

The point-to-point to reviewersâ comments are listed below.

Reviewer #1 (Remarks to the Author):

The manuscript reports the discovery and development of an inhibitor of thymidylate kinase (TMPK) and cytidylate kinase (CMPK), JMF4073, aimed as second line therapeutics to cure TKI resistance chronic myeloid leukemia (CML) caused by T315I Bcr-Abl mutation. They found the drug resistance of the first line therapeutics comes from upregulation of GSH synthesis. However, JMF4073 is potent with wt-Bcr-Abl cells, but not T315I-Bcr-Abl cells they aimed at. Therefore, they proposed a combination therapy of JMF4073 with GSH synthesis inhibitor to treat T315I resistance CML. This work provides insight of T315I resistance CML and the compound JMF4073 is a potent TMPK and CMPK inhibitor. However, the project is not well designed and the manuscript is not well organized. It needs major revision for publication.

RE: We fully agree with this point.

The sequence of figures has been reorganized according to the logical development of this study.

ï- Figure 1 and Table 1 describe JMF4073 as a small compound that inhibits both CMPK and TMPK.

ï- Figure 2 shows that WT-Bcr-Abl leukemia cells are vulnerable to JMF4073, but T315I-Bcr-Abl leukemia cells are less susceptible. The toxicity of JMF4073 to myeloid progenitors is lower as compared to other nucleoside analogs.

ï- Figure 3-4 aims to understand what causes the differential susceptibility. Data in these two figures provide evidence that the activation of ATF4-mediated transcriptional network in T315I-Bcr-Abl-transformed myeloid cells leads to the increases in dTTP pool and GSH level that prevent replication stress observed in WT-Bcr-Abl

ï- Figure 5 demonstrates that it is the upregulation of GSH, but not dTTP biosynthesis, that dampens JMF4073 susceptibility in T315I-Bcr-Abl cells.

ï- Figure 6 shows that reducing GSH synthesis in combination with JMF4073 depletes dTTP/dCTP pools to eradicate T315I-Bcr-Abl leukemia.

ï- Figure 7 strengthens our notion that JMF4073 sensitivity is inversely correlation with ATF4 and GSH level in different human CML-BC cell lines. There is a synergistic synthetic lethality by the combinatory treatment of GSH reducing agent in combination with JMF4073 in human CML-BC cell lines.

Major points,

1. The project design and logical in the study is not clear. Basically, the manuscript is difficulty to read and understand. They need to have a clear description of their goals and logical at the beginning of the manuscript, such as in the Abstract and Introduction.

RE: Thank you so much for pointing out the major weakness of this manuscript. All figures have been reorganized according to the initial rationale and experimental designs for the questions that arose from the results during developing this study. We hope the revision is comprehensible and satisfactory for understanding.

2. The title is confusing to start with. The relationship of Glutathione, Bcr-Abl-induced Replication Stress and Vulnerability, and Inhibitor of Pyrimidylate Kinases are confusing or misleading.

RE: We have changed the title to "Glutathione Determines Chronic Myeloid Leukemia Vulnerability to an Inhibitor of CMPK and TMPK" to fit the context of this study.

3. Abstract, it is not clear that this work is aimed at a metabolic study to understand T315I resistance CML or a drug discovery. Need to revise with a better logical flow.

RE: We have revised the abstract and deleted "a metabolic study". Following the limit of 150 words, we hope the logical flow of the revised abstract is improved.

4. Abstract. There are many confusions in the wording. Here is an example, "T315I mutation to acquire resistance to non-toxic tyrosine kinase inhibitors (TKI)" how about toxic TKI? All TKI are non-toxic or only resistance to non-toxic TKI?

RE: We have removed the confusing term, "non-toxic", in the revision.

5. Introduction. Need to provide more backgrounds for general readers and use general terms as much as possible. For example, a brief description of t(9;22) would be helpful. Some parts of the Results and Discussion should be moved to introduction to prepare readers early to understand the work. For example, some the first paragraph of Discussion provided the Aims of this work and should be moved to Introduction. Some concepts popped up in the Results such as "Timeless/Tipin with replisome" (line 97), "Slc1a1 and Slc7a" (line 108) should be briefly described in Introduction.

RE: Thank you for this suggestion. We add the description of t(9;22) translocation for Bcr-Abl fusion gene in CML, and reorganized the description in the introduction, results, and discussion.

6. Introduction needs to be reorganized. Brief description of the results should be put at the end of the introduction, instead of in every paragraph.

RE: We have added a brief description of overall results at the end of the introduction.

7. In the results, for up or down regulation, should indicate clearly what it refers to and compares with. For example, line 135-137, Fig3c, PSAT and SHMT2 level in 32D also should be shown to compare.

RE: Data of PSAT1 and SHMT2 RNA level in 32D are shown in Figure 4c.

8. Results. "Altogether, these data suggest that upregulation of GSH rather than dTTP pool is responsible for avoiding oncogene-induced replication stress in T315I-Bcr-Abl-32D cells." (line 145) it is not very clear why draws this conclusion. Need more explanations.

RE: Through SHMT2 knockdown to reduce dTTP pool in T315I-Bcr-Abl-32D cells (Figure 4g), we did not find the induction of replication stress nor the change in JMF4073 sensitivity (Figure 5a and b). As a contrast, reducing GSH level by BSO or UK5099 leads to GSH reduction that increases JMF4073 sensitivity in T315I-Bcr-Abl-32D cells with a concurrent induction of replication stress (Figure 5 c-e, Figure 6 a-c). Conversely, increasing GSH in WT-Bcr-Abl cells by NAC incubation abolishes replication stress and causes JMF4073 insensitivity (Figure 5 f-h). These results suggest that it is GSH level rather than dTTP level that determines JMF4073 sensitivity in CML cells.

9. Results. Some statements lack of reference. For example, (line 77) Compelling evidence has suggested that oncogene-induced replication stress is a driving force in cancer progression. "need references."

RE: Ref is added in 30 and 31 .

10. Discussion. Some interesting questions could be discussed here. For example, why only dTTP and dCTP pools were affected, but not purine pools? (Figure 2&4 can be used for answers). Also, limitation of the work should be discussed. For example, JMF4073 is failed as an T315I resistance CML therapeutic. It works only together with GSH inhibitors. At least three of GSH inhibitors are tried in this work, from BSO to UK-5099 to erastin, with increasing potency, which surely will lead to increased toxicity. As a prove of concept, this going-around approach really works? Or should think about

designing a better inhibitor directly work at T315I BCR-ABL?

RE: Agree with this point. In the revised version, Data of Figure 4 provide evidence that the higher level of ATF4 expression mediates the upregulation of SHMT2 for dTTP synthesis and a number of amino acid transporter genes for GSH biosynthesis in T315I-Bcr-Abl. SHMT2 function generates the production of CH₂-THF, which gives one-carbon for dTMP synthesis from dUMP (Figure 4b and g). The data explain why dTTP level is so different in WT- and T315I-Bcr-Abl. We have added this mechanistic explanation in the discussion. A direct inhibitor, Ponatinib, for T315I-Bcr-Abl is already available. However, as stated, Ponatinib exerts unwanted toxic effect. In this manuscript, we demonstrated that Ponatinib treatment also suppressed general tyrosine phosphorylation in 32D cells without Bcr-Abl transformation, indicating its non-specific effect (FIG 2a). Of course, a direct inhibitor would be a much better and ideal approach. However, a specific inhibitor against T315I-Bcr-abl remains unavailable. Our data showing that a drastic synergistic effect of JMF4073 and GSH inhibitor such as erastin on eradicating CML blast-crisis cells suggests this combination can be an alternative choice before the birth of a specific inhibitor of T315I-Bcr-Abl. Moreover, a number of TKI-resistant CMLs unnecessarily involve the acquisition of T315I mutation. Therefore, a new combinatory therapy is still another option for therapy.

Minor points,

1. Many self-invented terms were used, which made the manuscript hard to read. For example, what is "dNTPs blockers" (line 28) and "nucleotide blockers" (line 63)? Do they mean compounds with nucleotide-like structure? If so, please define these terms.

RE: We change the wording.

2. Pyrimidylate Kinase is not a general term. Please make definition.

RE: We changed pyrimidylate kinase to Thymidylate and cytidylate kinases (TMPK and CMPK).

3. Need to work on English for the proper descriptions and words.

RE: English Editing has been done.

Reviewer #2 (Remarks to the Author):

The authors explored therapeutic application of targeting GSH metabolism in treating chronic myeloid leukemia cells upon inhibiting pyrimidylate 30 kinases (TMPK and CMPK) via a homemade drug JMF4073. Overall, the study presented an interesting findings that CML cells containing low levels of glutathione require dNTPs supply to overcome DNA damage and replication stress for survival. Blocking glutathione in combination with JMF4073 led to synthetic lethality in WT-Bcr-Abl-32D. There are only several questions needed to be addressed before publication.

1, How exactly GSH level regulated replication stress in leukemia cells? High GSH and NAC suppressed replication stress. However, GSH and NAC are wellknown antioxidants.

So, the role of ROS should also be examined in the context of study.

RE: Data in Figure 3c and d showed ROS and GSSG measurement in 32D, WT- and T315I-Bcr-Abl-32D cells. ROS level is higher in T315I-Bcr-Abl cells than that in WT-Bcr-Abl- cells, consistent with the increase in GSSG level. However, the total amount of reduced GSH is indeed higher in T315I-Bcr-Abl cells. Of note, GSSG only accounts for a small proportion of total level of GSH (FIG 3e). Thus, higher GSH level in T315I-Bcr-Abl indeed buffers ROS production, by which replication stress is prevented (We discussed in Page 14-15).

2, The paper showed that wt Bcr-Abl might regulated enzymes invovled in the biosynthesis of dNTPs, such as SHMT2. What's the potential mechanisms?

RE: In this revision, we provide evidence that ATF4 is highly activated in T315I-Bcr-Abl-32D cells. Activation of ATF4 mediates the integrated stress network shown in Figure 4, which leads to upregulation of SHMT2 transcription. The function of SHMT2 is to generate 5,10-CH₂-THF for dTMP formation from dUMP. As a result, dTTP pool is expanded in T315I-Bcr-Abl-32D cells. Knockdown of SHMT2 reduces dTTP pool in T315I-Bcr-Abl-32D cells. Therefore,

the mechanism for SHMT2-mediated increase of dTTP pool is due to higher activation of ATF4 in T315I-Bcr-Abl-32D cells. We explain these results in results and discussion (The first paragraph on page 9-10 and on page14).

Of note, ATF4 knockdown reduces the total GSH level. Since ATF4-mediated network also upregulates amino acid transporters such as SLC1A1 and SLC7A8 and the production of glycine from SHMT2 reaction for GSH biosynthesis, it is likely that ATF4-mediated integrated stress responses act in concert to increase GSH biosynthesis.

We wish that the clarity of this revision is improved for reading and meet the criteria for publication in Communication Biology. We hope that the findings of this study are of interest to the readers for future exploitation in cancer therapy.

Sincerely,
Zee-Fen

Zee-Fen Chang, Ph.D., Chair Professor,
Institute of Molecular Medicine, College of Medicine,
National Taiwan University,
Taipei, Taiwan

Version 1:

Reviewer comments:

Reviewer #1

(Remarks to the Author)

The authors responded to all comments.

Reviewer #2

(Remarks to the Author)

The reviewer has no further questions regarding the current manuscript.
